# Antibiotic Resistance of *Helicobacter pylori* and Related Risk Factors in Yangzhou, China: A Cross-Sectional Study

**DOI:** 10.3390/jcm12030816

**Published:** 2023-01-19

**Authors:** Yun Zhang, Xinyi Feng, Lijun Bian, Yan Zhang, Qian Li, Yemin Xu, Qiang She, Caiwang Yan, Guotao Lu, Jian Wu, Weiming Xiao, Yanbing Ding, Bin Deng

**Affiliations:** 1Department of Gastroenterology, The Affiliated Hospital of Yangzhou University, No. 368 Hanjiang Middle Road, Yangzhou 225001, China; 2Department of Emergency, The Affiliated Suqian Hospital of Xuzhou Medical University, Suqian 223800, China; 3Department of Gastroenterology, Wuzhong People’s Hospital of Suzhou, Suzhou 215000, China; 4Department of Epidemiology, Center for Global Health, School of Public Health, Nanjing Medical University, Nanjing 211166, China

**Keywords:** *Helicobacter pylori*, antibiotic resistance, gene mutation, risk factors

## Abstract

Background: The antibiotic resistance of *Helicobacter pylori* (*H. pylori*) is a common cause of treatment failure. Previous studies showed that *H. pylori* resistance may be related to some characteristics of patients. This study intended to investigate the resistance of *H. pylori* to five commonly used antibiotics and risk factors in Yangzhou, China. Methods: We recruited the subjects who joined the endoscopic screening program organized by the Affiliated Hospital of Yangzhou University between April 2018 and September 2019 and endoscopists would take biopsy samples from the antrum and the corpus of the stomach. The antrum biopsy specimens were used to culture *H. pylori*. Next, we extracted DNA from *H. pylori* strains and performed the specific DNA amplification. Finally, we use gene chip technology to test the susceptibility to clarithromycin, levofloxacin, metronidazole, amoxicillin and tetracycline. Multivariate logistic analyses were also performed to determine the risk factors for antibiotic resistance of *H. pylori*. Results: A total of 461 *H. pylori* strains were finally collected. The resistance rate of *H. pylori* to clarithromycin, levofloxacin, metronidazole, amoxicillin and tetracycline was 41.0%, 44.9%, 38.8%, 6.3% and 1.1%, respectively. In addition, 16 multi-resistance patterns were detected, and strains resistant to all five antibiotics were not found. Multivariate analysis showed that past medical history and clinical outcomes were significantly associated with the resistance to clarithromycin. Drinking, gastrointestinal symptoms and a family history of gastric cancer were significantly associated with the resistance of *H. pylori* to levofloxacin. Especially gastrointestinal symptoms were significantly associated with the resistance of *H. pylori* to any antibiotic. Conclusion: The resistance rates of *H. pylori* to clarithromycin, levofloxacin and metronidazole were very high in Yangzhou, China, various factors were related to bacterial resistance, and grasping these influencing factors can guide treatment.

## 1. Introduction

*Helicobacter pylori* (*H. pylori*) infection is probably one of the most prevalent “infectious diseases” in the world, with approximately 50% or more of the world’s population having *H. pylori* in their stomachs [1]. In China, there is also a heavy burden of *H. pylori* infection. A new meta-analysis showed that the overall prevalence of *H. pylori* infection in mainland China was 44.2% [2]. It is generally considered that *H. pylori* infection is associated with chronic gastritis, peptic ulcers, lymphoid tissue lymphoma, and gastric cancer [3,4]. The successful eradication of *H. pylori* is of great significance for curing chronic gastritis and reducing the incidence of gastric cancer [5,6].

The eradication regimens of *H. pylori* have changed from empirical triple therapies to bismuth quadruple therapies, but there still exists eradication failure [7]. Previous studies have shown that the primary reasons for the failure of *H. pylori* eradication included poor patient compliance, *H. pylori*-related factors, host factors, environmental factors and recurrence [8]. One of the most important reasons for eradication failure is antibiotic resistance of *H. pylori*. It was estimated that from 2010–2016, *H. pylori* resistance rates in China to clarithromycin, metronidazole, and levofloxacin were 37%, 77%, and 33% [9]. Meanwhile, the antibiotic resistance of *H. pylori* exhibited regional differences and appeared to be changing over time [10,11]. Therefore, screening for antibiotic resistance of *H. pylori* based on the local population and finding related risk factors can greatly help improve the effect of *H. pylori* eradication.

This study aimed to assess the antibiotic resistance situation of *H. pylori* strains and risk factors for antibiotic resistance of *H. pylori* in Yangzhou, China, which provide references for precise treatment of *H. pylori*.

## 2. Materials and Methods

### 2.1. Patients

This study collected the subjects from the endoscopic screening program organized by the Affiliated Hospital of Yangzhou University between April 2018 and September 2019. Inclusion criteria were: permanent residents over 18 years old of Yangzhou, cases with a positive result of ^13^C-urea breath test (^13^C-UBT) and isolating and culturing *H. pylori*, cases with complete questionnaires, endoscopies and pathology reports. Exclusion criteria were: cases complicated with serious diseases (such as chronic vital organ failure and autoimmune diseases), cases complicated with digestive symptoms including hematemesis, abdominal pain and black stool, and cases with difficulty in completing gastroscopies.

### 2.2. Data Collection

Epidemiological data were obtained by the clinical medicine students of Yangzhou University, including general demographic data, life habits, history of stomach disease, and family tumor history. Body mass index (BMI) was calculated as weight divided by height squared (kg/m^2^). Current smoking was defined as people who still smoke at least one cigarette a day and had been smoking for more than half a year or had quit smoking for less than 15 years, otherwise, they were considered non-smokers. Current drinking was defined as people who drank more than once a week for more than one year, otherwise, they were considered non-drinkers. Gastrointestinal symptoms were defined as that people had abdominal pain, anorexia, fullness, heartburn, dysphagia, and so on.

### 2.3. Isolation of H. pylori Strains

All participants underwent gastroscopies by experienced endoscopists who performed more than 1000 endoscopies annually. The endoscopists took two biopsy samples from the antrum and one biopsy sample from the corpus. One sample from the antrum was used for *H. pylori* culture. Another biopsy sample from the antrum and the biopsy sample from the corpus were used for histopathological investigations.

The specimens from the antrum of the stomach were inoculated onto Columbia agar plates (Thermo Fisher Scientific, Waltham, MA, USA) with an antibiotic selective supplement containing 10% fetal bovine serum (Sijiqing Bioengineering Materials Company, Hangzhou, China), 5 mg/L trimethoprim (Sangon Biotech, Shanghai, China), 10 mg/L vancomycin (Sangon Biotech, Shanghai, China), and 0.38 mg/L polymyxin B (Sangon Biotech, Shanghai, China). They were incubated at 37 °C for 3–5 days under microaerophilic conditions containing 5% O_2_, 10% CO_2_, and 85% N_2_. *H. pylori* colonies were initially identified by their typical morphology (transparent, round, and moist), which were then transferred to a fresh Columbia blood agar plate for 48–72 h incubation. The isolates were frozen at −80 °C in a brain-heart infusion storage medium (Bo Wei Technology Company, Shanghai, China) containing 20% glycerol until assayed.

### 2.4. DNA Extraction from H. pylori Strains

Isolated strains were unfrozen at room temperature. Following the removal of the supernatant, the cell pellet was fully incubated in 200 μL of normal saline containing 20 μL of pathogen DNA extracting protease K. Nucleic acid extraction was performed according to instructions provided by the reagent manufacturer (Hangzhou Meilian Medical Inspection Company, China), and isolated DNA was stored at 4 °C.

### 2.5. Polymerase Chain Reaction (PCR)

Based on previous articles [12,13], we detected several point mutations of 23S rRNA, 16S rRNA, gyrA, rdxA, and PBP1 in *H. pylori*, which can be found in Table 1. The specific DNA amplification was performed as follows (primer pairs were designed by Hangzhou Meilian Medical Inspection Company, according to the sequence of *H. pylori* UreA gene, VacA gene, and five drug-resistant genes). Uracil-DNA glycosylase enzyme reaction was performed at 50 °C for 10 min, followed by pre-denaturation for 10 min at 95 °C, then 45 cycles of denaturation at 95 °C for 30 s, extension at 56 °C for 30 s, and a further extension for 30 s at 72 °C. Finally, A primer extension for 5 min at 72 °C was applied, and samples were stored at 4 °C for use.

### 2.6. Gene Chip Detection of H. pylori

*H. pylori*-associated gene probes distributed on membrane strips are shown in Figure 1. The membranes were put into the 24-well plate sequentially, in which 1 mL of liquid A (100 mL 20 × SSC, 10 mL 10% sodium dodecyl sulfonate (SDS) plus pure water to 1000 mL) was added. After preheating in boiling water for 20 min, the PCR products were added and hybridized at 48 °C for 1.5 h. Next, the reaction solution was discarded, and membranes were gently mixed in 1 mL of pre-warmed liquid B (25 mL 20 × SSC, 10 mL 10% SDS plus pure water to 1000 mL) at 48 °C for 15 min, which were later gently mixed in the incubation solution at room temperature for 30 min, and liquid A for 5 min twice. After washing in liquid C (100 mL 1 M sodium citrate plus pure water to 1000 mL) at room temperature for 1–2 min, membranes were infiltrated in the chromogenic solution (19 mL liquid C added to 1 mL of 3,3’,5,5’-tetramethylbenzidine (TMB) and 2 μL of 30% H_2_O_2_) in the dark at room temperature for 10 min. They were finally rinsed with pure water to observe the results.

### 2.7. Statistical Analysis

All statistical analyses were performed using SPSS statistical software package version 19.0 (SPSS Inc, Chicago, IL, USA). Antibiotic resistance rates of *H. pylori* isolates were expressed as frequencies and percentages. Categorical variables were compared using the Fisher’s exact test or χ^2^ test. Multivariable logistic regression was used to examine the possible predictors of antibiotic resistance of *H. pylori* and variables who were screened out from the univariate regression analysis with *p* value < 0.20 were included in this model. Differences with a two-tailed *p* value < 0.05 were considered statistically significant.

## 3. Results

### 3.1. Baseline Information of Participants

461 individuals were included in this study (Table 2). They were 192 (41.6%) males and 269 (58.4%) females aged from 32 to 75 years, with a mean age of 56.3 ± 8.3 years. The BMI of the included subjects ranged from 15.2 to 39.5 kg/m^2^, with a mean value of 24.2 ± 3.2 kg/m^2^. There were 132 (28.6%) smokers and 78 (16.9%) drinkers. In clinical characteristics, 104 (22.6%) cases had gastrointestinal symptoms, and 51 (11.1%) cases had ulcer or cancer under the endoscopies. In 87 (18.9%) cases, patients had a history of superficial gastritis, and in 69 (15.0%) cases, their first-degree relatives had gastric cancer ever.

### 3.2. Antibiotic Resistance Patterns of H. pylori

Among the 461 isolated *H. pylori* strains, 361 (78.3%) were resistant to at least one tested antibiotic. The resistance rate of *H. pylori* strains to levofloxacin (LVX) was the highest (44.9%, 207/461), followed by clarithromycin (CLR) (41.0%, 189/461) and metronidazole (MTZ) (38.8%, 179/461), while that of amoxicillin (AMX) (6.3%, 29/461) and tetracycline (TET) (1.1%, 5/461) was lower than 10% (Figure 2).

The distribution of the *H. pylori*-resistant gene locus is shown in Table 3. The *H. pylori* resistance gene locus by the gene chip identified CLR resistance to be mainly at A2143G, accounting for 98.4% (186/189). Four mutation sites of gyrA gene were detected in 207 LVX-resistant strains, the most common was N87K, accounting for 63.3% (131/207), followed by D91N (14.5%, 30/207), D91G (14.5%, 30/207) and D91Y (7.7%, 16/207). MTZ resistance was mainly due to mutation at G616A of rdxA gene (100.0%, 179/179). Three mutation sites of PBP1 gene were detected in 29 AMX resistant strains, namely T556S (62.1%, 18/29), N562Y (34.5%, 10/29) and N562D (3.4%, 1/29). All five TET resistant strains had mutations in the AGA926-928TTC site of 16S rRNA gene (100.0%, 5/5).

Antibiotic resistance patterns of *H. pylori* are shown in Table 4. Only 21.7% (100/461) of the isolated strains were sensitive to all five antibiotics. Among the other 361 strains, 42.3% (195/461) were resistant to more than one antibiotic, including 31.5% (145/461) resistant to two antibiotics, 10.2% (47/461) resistant to three, and 0.7% (3/461) resistant to four. In total, 16 multi-resistance patterns of *H. pylori* were identified. Among the dual-resistance patterns, CLR + LVX (14.3%, 66/461), LVX + MTZ (9.8%, 45/461), and CLR + AMX (5.9%, 27/461) were the top combinations. The main triple-resistance patterns were CLR + LVX + MTZ (6.7%, 31/461) and CLR + LVX + AMX (2.6%, 12/461).

### 3.3. Risk Factors Associated with Antibiotic Resistance of H. pylori

We did not analyze risk factors associated with AMX and TET resistance because of the low resistance rates. Potential factors associated with antibiotic resistance of *H. pylori* obtained from the univariable analysis are summarized in Table 5. It was shown that gastrointestinal symptoms, history of superficial gastritis and endoscopic findings (ulcer or cancer) were significantly associated with the resistance of *H. pylori* to CLR. Strains isolated from people who drunk regularly were more commonly resistant to LVX. Moreover, gastrointestinal symptoms, previous history of superficial gastritis and first-degree relatives with gastric cancer were significantly associated with the resistance of *H. pylori* to LVX. The resistance to MTZ was found significantly frequent among subjects with gastrointestinal symptoms. The previous history of superficial gastritis was also associated with the resistance of *H. pylori* to MTZ. Collectively, alcohol consumption, presence of gastrointestinal symptoms, history of chronic superficial gastritis, and family history of gastric cancer in first-degree relatives were considered as risk factors associated with the resistance of *H. pylori* to CLR, LVX and MTZ we tested, whilst sex and age were not correlated with them.

Furthermore, multivariate logistic analysis showed that patients with a history of superficial gastritis had a higher likelihood of clarithromycin resistance (*p* = 0.021, OR: 1.74, 95%CI 1.09–2.79). People with ulcers or cancers detected by gastroscopy had a lower risk of clarithromycin resistance (*p* = 0.030, OR: 0.48, 95%CI 0.25–0.93). With regard to levofloxacin, the results showed that former drinkers (*p* = 0.005, OR: 0.47, 95%CI 0.28–0.80) and patients with a family history of first-degree relatives with gastric cancer (*p* = 0.040, OR: 0.56, 95%CI 0.33–0.97) were less likely to develop resistance, while those with gastrointestinal symptoms were more likely to develop resistance than those without symptoms (*p* = 0.029, OR: 1.65, 95%CI 1.05–2.57). Especially gastrointestinal symptoms were significantly associated with the resistance of *H. pylori* to any antibiotics (*p* = 0.043, OR 1.93, 95%CI 1.05–3.52), and the correlation still existed after adjusting sex, age, gastrointestinal symptoms, history of superficial gastritis, and family history of gastric cancer in first-degree relatives (Table 6).

## 4. Discussion

Yangzhou is geographically located in the middle of Jiangsu province, China, where the incidence of gastric cancer remains high, with 44.05/100,000 in 2013 [14]. *H. pylori* infection is regarded as the major risk factor for gastric cancer. A long-term cohort study in Taiwan showed that eradicating *H. pylori* could remarkably reduce gastric cancer incidence and mortality [15]. Unfortunately, the high prevalence of *H. pylori* infection and the increasing antibiotic resistance constitute the main challenges for current treatment. Through the previous follow-up investigation, we have found a disturbing phenomenon of *H. pylori* eradication treatment failure in a large percentage of the population. Therefore, a local resistance analysis is urgently required.

Currently, we usually obtain information about antibiotic resistance of bacteria through *H. pylori* culture or molecular biological detection. Compared with the agar dilution method (E test) which is generally regarded as the gold standard of antibiotic susceptibility testing, the molecular techniques also show excellent specificity and sensitivity [16,17]. In addition, although DNA sequencing seems to be more convincing in detecting drug resistance, we still need to develop more straightforward and more accessible methods than sequencing to detect mutations for better clinical practice. In this study, gene chip technology was used to detect *H. pylori* resistance. The principle of DNA chip is the method of hybridization and sequencing, and we can quickly obtain the gene sequence of the tested DNA fragments by hybridization with a set of DNA probes with known sequences. The gene chip technology is a suitable multi-target, multi-site methodology that can quickly detect and characterize *H. pylori* infection and mutation of multiple drug resistant sites.

First-line eradication treatment is essential in China because the rate of *H. pylori* reinfection after successful treatment is low (1.5% per person-year) [18], while the global annual reinfection rate of *H. pylori* is estimated at 3.1% [19]. It is suggested that effective therapy is necessary to improve the first-line eradication rate of *H. pylori*. Antibiotic resistance rate of *H. pylori* varies among countries or regions. In Italy, resistance rate to CLR was 35.9% in 2016 [20], which was 43.7% in Korea [21]. The present study showed that the resistance of *H. pylori* to CLR (41.0%) was slightly higher than that in Zhuanghe (31%) [22]. The significantly higher resistance rate of *H. pylori* to CLR in Yangzhou, China might be attributed to the emergence of drug-resistant bacteria caused by long-term and broad application of antibiotics. In addition, in this study, the most common mutation site for clarithromycin resistance was A2143G, similar to Bachir’s results [23]. According to consensus recommendations, LVX-containing regimen was generally not suggested as an initial treatment, which was preferred as an alternative for rescue therapy because of the high rate of drug resistance [24]. The resistance rate of *H. pylori* to LVX was 38.8% in Taiwan, China in 2019 [25] and 56% in Zhuanghe, China in 2019 an area of high risk of gastric cancer [22]. In the present study, the resistance rate of *H. pylori* to LVX remained the highest among the tested bacteria, which the extensive use of quinolones for respiratory and urogenital infections may explain. Meanwhile, *H. pylori* resistance to LVX was caused by mutations at sites 87 and 91 of the GyrA gene. In this study, the mutation rates of 87 and 91 loci were found to be 63.3% and 36.7%, respectively, which was consistent with a prior report [26]. MTZ is a 5-nitroimidazole drug that is widely used for general anaerobic infections worldwide. MTZ resistance was primarily due to mutation at G616A. The overall drug resistance rate of MTZ in this study was lower than the average rate (61%) in China reported in a meta-analysis [27], but this still needs our high attention. Several studies showed that increasing the dosage and frequency of MTZ could reduce high-level resistance to MTZ [28,29], but this could increase the incidence of adverse events [29].

Our study showed a low resistance rate of *H. pylori* to AMZ, which was consistent with worldwide data. The present study found a PBP1 gene mutation mainly caused AMX resistance at T556S (62.1%), N562Y (34.5%) and N562D (3.4%). Nowadays, AMZ-containing bismuth quadruple therapy is always used as the preferred choice for *H. pylori* eradication treatment. In addition, high-dose dual therapy has been well concerned because of its great efficacy and fewer adverse events. Song et al. reported that dual-therapy consisting of esomeprazole and amoxicillin four times daily was not inferior to, and even superior to triple-therapy plus bismuth therapy as first-line *H. pylori* eradication treatment [30]. However, these findings were not yet consistent and remained to be confirmed with further studies. *H. pylori* resistance to TET was mainly due to a substitution mutation at the locus of 16s rRNA gene AGA926-928TTC [31]. TET resistance rate was low all over the world and was recommended to replace AMX for people allergic to penicillin by experts.

Our study showed that 42.3% of *H. pylori* strains were resistant to at least two antibiotics, with the main resistance patterns being CLR + LVX (14.3%), CLR + MTZ (9.8%), and CLR + LVX + MTZ (6.7%). This suggested that we should use the combinations mentioned above of antibiotics with caution for eradication therapy. In addition, we also identified several quadruple-resistance patterns, and no strains were resistant to all five tested antibiotics. The analysis of multi-drug resistance data told us that if bacterial resistance was determined before treatment, the probability of successful eradication would definitely increase.

Multivariate analysis data showed that a history of chronic superficial gastritis was associated with the resistance of *H. pylori* to CLR. *H. pylori* is the major cause of chronic gastritis, and a long-term *H. pylori* infection and the formation of bacterial biofilms may lead to antibiotic resistance [32], and intervention at the early stage of the disease is a wise choice. Subjects with an endoscopic diagnosis of peptic ulcer or cancer presented a lower risk of resistance to CLR, which was consistent with a previous study in France [33]. The resistance of *H. pylori* to LVX was significantly associated with alcohol consumption. In clinical practice, LVX is usually not prescribed to drinkers with concerning the disulfiram-like reaction, which might contribute to a low resistance rate of LVX among drinkers. Furthermore, we have found that people with a family history of first-degree relatives with gastric cancer had a lower possibility of resistance to LVX, and the potential cause is unclear. Gastrointestinal symptoms were significantly associated with LVX and any antibiotic resistance. We considered that people with gastrointestinal symptoms had already been infected by *H. pylori* but they always ignored this because they did not pay attention to their health. Clinicians must strengthen health education.

Some limitations in the present study should be noted. Firstly, we performed the endoscopic screening program in two small towns in Yangzhou, China, and the sample size was small. We intend to expand the crowd in the future to increase the credibility of conclusions. Secondly, we detected known mutation sites to determine antibiotic resistance of *H. pylori* using microarray technology, which might cause an underestimation of antibiotic resistance. In addition, a follow-up investigation on *H. pylori* eradication in positive populations had yet to be lacking, which would be explored in the future. Finally, our study was based on gene chip detection and attention should be paid to the bacterial culture for sensitivity tests at the same time in the future.

## 5. Conclusions

The resistance rate of *H. pylori* to CLR, LVX and MTZ remained high in Yangzhou, China, and serious multi-drug resistance cases were detected in this region. In order to achieve a >90% eradication rate of *H. pylori* in first-line treatment, a combination medication of the above three antibiotics should be avoided in local people from Yangzhou, China. In addition, the history of chronic superficial gastritis and a family history of gastric cancer in first-degree relatives were the risk factors for the resistance to clarithromycin and levofloxacin, respectively. The gastrointestinal symptom was a risk factor for resistance to any antibiotic. In the follow-up screening activities, we should strengthen the monitoring of drug resistance of *H. pylori*, carry out drug resistance detection on the high-risk population of drug resistance, avoid the preference of certain antibiotics, and increase the health publicity for local residents to maintain a good lifestyle and seek timely treatment of stomach discomfort symptoms.

## Figures and Tables

**Figure 1 jcm-12-00816-f001:**
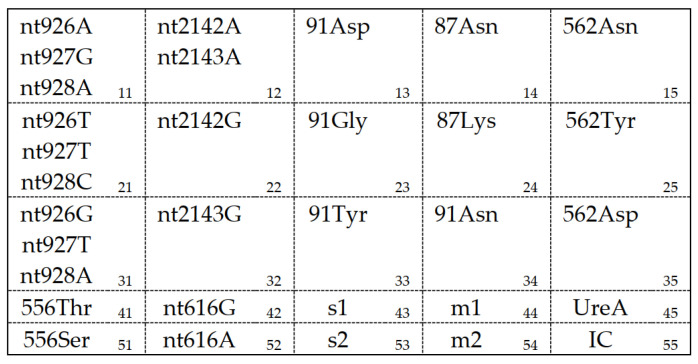
The distribution of *H. pylori*-associated gene probes on membrane strips. 11–15, 41–42 represent *H. pylori* wild–type sequences; 21–25, 31–35, 51–52 represent *H. pylori* mutant sequences; 43–44, 53–54 represent *H. pylori* virulence genotyping.

**Figure 2 jcm-12-00816-f002:**
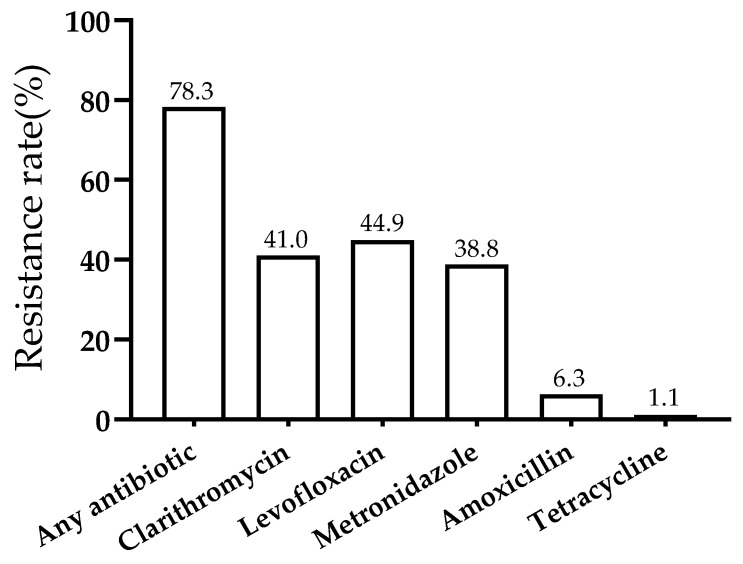
The resistance rate of *H. pylori* strains to Clarithromycin, Levofloxacin, Metronidazole, Amoxicillin, Tetracycline.

**Table 1 jcm-12-00816-t001:** Point mutations of *H. pylori* genes by gene chip detection.

Antibiotic	Gene	Mutation Site
Clarithromycin	23S rRNA	2142, 2143
Levofloxacin	gyrA	87, 91
Metronidazole	rdxA	616
Amoxicillin	PBP1	556, 562
Tetracycline	16S rRNA	926–928, 926–927

**Table 2 jcm-12-00816-t002:** The baseline information of *H. pylori* isolates (N = 461).

Factors	*n*	%
Sex (Female)	269	58.4
Age, years (≥60)	179	38.8
BMI, kg/m^2^ (≥24)	226	49.0
Smoking (Current smoking)	132	28.6
Drinking (Current drinking)	78	16.9
Gastrointestinal symptoms (Yes)	104	22.6
History of superficial gastritis (Yes)	87	18.9
First-degree relatives with gastric cancer (Yes)	69	15.0
Endoscopic findings (Ulcer or cancer)	51	11.1

BMI, body mass index.

**Table 3 jcm-12-00816-t003:** Detection of *H. pylori* resistant gene locus by gene chip technology.

Antibiotics	Resistant Sites	Detection Number	
		*n*	%
Clarithromycin	A2143G	186	98.4
	A2142G	3	1.6
Levofloxacin	N87K	131	63.3
	D91N	30	14.5
	D91G	30	14.5
	D91Y	16	7.7
Metronidazole	G616A	179	100
Amoxicillin	T556S	18	62.1
	N562Y	10	34.5
	N562D	1	3.4
Tetracycline	AGA926-928TTC	5	100
	AG926-927GT	0	0

**Table 4 jcm-12-00816-t004:** Antibiotic resistance patterns of *H. pylori* strains (N = 461).

Susceptibility Test Results	No. of Strains (*n*)	Resistance Rate (%)
CLR	47	10.2
LVX	45	9.8
MTZ	69	15.0
AMX	5	1.1
TET	0	0
CLR + LVX	66	14.3
LVX + MTZ	45	9.8
CLR + MTZ	27	5.9
CLR + AMX	1	0.2
LVX + AMX	3	0.7
LVX + TET	1	0.2
MTZ + AMX	1	0.2
AMX + TET	1	0.2
CLR + LVX + MTZ	31	6.7
CLR + LVX + AMX	12	2.6
CLR + MTZ + AMX	1	0.2
CLR + MTZ + TET	1	0.2
LVX + MTZ + AMX	1	0.2
LVX + AMX + TET	1	0.2
CLR + LVX + MTZ + AMX	2	0.4
CLR + MTZ + AMX + TET	1	0.2

CLR, clarithromycin; LVX, levofloxacin; MTZ, metronidazole; AMX, amoxicillin; TET, tetracycline.

**Table 5 jcm-12-00816-t005:** Factors associated with the resistance of *H. pylori* to clarithromycin, levofloxacin, metronidazole and any antibiotic.

Factors	Clarithromycin		*p*	Levofloxacin		*p*	Metronidazole		*p*	Any Antibiotic		*p*
	S (*n* = 272)	R (*n* = 189)		S (*n* = 254)	R (*n* = 207)		S (*n* = 282)	R (*n* = 179)		S *(n* = 100)	R (*n* = 361)	
Sex			0.365			0.674			0.291			0.936
Male	118	74		108	84		112	80		42	150	
Female	154	115		146	123		170	99		58	211	
Age, years			0.612			0.401			0.202			0.615
<60	169	113		151	131		166	116		59	223	
≥60	103	76		103	76		116	63		41	138	
BMI, kg/m^2^			0.754			0.110			0.738			0.996
<24	137	98		138	97		142	93		51	184	
≥24	135	91		116	110		140	86		49	177	
Smoking			0.388			0.638			0.875			0.275
No smoking	190	139		179	150		202	127		67	262	
Current smoking	82	50		75	57		80	52		33	99	
Drinking			0.805			0.003			0.942			0.219
No drinking	47	31		199	184		48	30		21	57	
Current drinking	225	158		55	23		234	149		79	304	
Gastrointestinal symptoms			0.034			0.021			0.752			0.041
No	220	137		207	150		217	140		85	272	
Yes	52	52		47	57		65	39		15	89	
History of superficial gastritis			0.012			0.097			0.158			0.159
No	231	143		213	161		223	151		86	288	
Yes	41	46		41	46		59	28		14	73	
First-degree relatives with gastric cancer			0.383			0.036			0.746			0.111
No	228	164		208	184		241	151		80	312	
Yes	44	25		46	23		41	28		20	49	
Endoscopic findings			0.017			0.531			0.807			0.736
Others	234	176		228	182		250	160		88	322	
Ulcer or cancer	38	13		26	25		32	19		12	39	

S, Sensitivity; R, Resistance.

**Table 6 jcm-12-00816-t006:** Factors associated with the resistance of *H. pylori* to clarithromycin, levofloxacin, metronidazole or any antibiotics using multivariable logistic regression.

Factors	Clarithromycin		Levofloxacin		Metronidazole		Any Resistance	
	*p*	OR(95%CI)	*p*	OR(95%CI)	*p*	OR(95%CI)	*p*	OR(95%CI)
Sex								
Sex (Female vs. Male)	0.477	1.15 (0.78–1.69)	0.165	0.73 (0.48–1.14)	0.329	0.83 (0.56–1.21)	0.968	0.98 (0.62–1.55)
Age, years								
≥60 (vs. <60)	0.428	1.17 (0.79–1.73)	0.928	0.93 (0.63–1.37)	0.139	0.74 (0.50–1.10)	0.757	0.95 (0.60–1.51)
BMI, kg/m^2^								
≥24 (vs. <24)		-		-		-		-
Drinking								
Current drinking (vs. No drinking)		-	0.005	0.47 (0.28–0.80)		-		-
Gastrointestinal symptoms								
Yes (vs. No)		-	0.029	1.65 (1.05–2.57)		-	0.043	1.93 (1.05–3.52)
History of superficial gastritis								
Yes (vs. No)	0.021	1.74 (1.09–2.79)		-		-		-
First-degree relatives with gastric cancer								
Yes (vs. No)		-	0.040	0.56 (0.33–0.97)		-		-
Endoscopic findings								
Ulcer or cancer (vs. Others)	0.030	0.48 (0.25–0.93)		-		-		-

## Data Availability

All data generated or analyzed throughout this research are included in this published article.

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
