# Peer review of "Antibiotic Resistance of Helicobacter pylori and Related Risk Factors in Yangzhou, China: A Cross-Sectional Study"

_jcm, 2023, doi:10.3390/jcm12030816_

Round 1
Reviewer 1 Report
Can be found in the attached pdf. Cannot find the results and discussion on gene probes.

Author Response
Dear Editors and Reviewers:
Thank you for your letter and for the reviewers’comments concerning our manuscript entitled “Identification of antibiotic resistant of Helicobacter pylori and related risk factors in Yangzhou, China: A cross-sectional study” (ID: jcm-2052575). Those comments are all valuable and very helpful for revising and improving our paper, as well as the important guiding significance to our researches. We have studied comments carefully and have made correction which we hope meet with approval. The main corrections in the paper and the responds to the reviewer’s comments are as flowing:
Responds to the reviewer’s comments:
Reviewer 1:
Comment 1: Were the 461 subjects all consecutive patients?
Response: The 461 subjects were not consecutive patients, we conducted screening for upper digestive tract diseases in rural areas of Yangzhou once a month as planned, with 300 to 500 people each time. This project helps us to better understand the health situation of the local people, and achieve early detection, diagnosis and treatment of early gastrointestinal cancer.
Comment 2: It is mentioned that the typical morphology of H. pylori is needle-like.
Response: In this paper, it is believed that the needle-like appearance of bacterial colonies is mainly due to our discovery in the process of bacterial culture and combined with the oral testimony of predecessors.
Changes in the text: We deleted the word.
Comment 3: Methods: problems of layout line 90.
Response: The primers used in the experiment were designed by Hangzhou Meilian Medical Inspection Company, and they designed a H. pylori genotype and drug resistance mutations detection kit.
Comment 4: Problems about the composition of main reagents.
Response: Thank you very much for your careful review. We apologize for the oversight.
Changes in the text: We note the main composition after each reagent.
Comment 5: Problems about the statistical methods.
Response: We appreciate your comments. In the process of data analysis, we consulted statisticians about the sample size of the study and which statistical method to choose. Currently, the EPV (events per variable) method is the most commonly used method for sample size calculation in regression analysis, that is, the number of events of each independent variable, in which events represent the category with a small number of dependent variables. The statistical simulation showed that the recommended empirical criterion in Logistic regression was EPV of at least 10, and the sample size was 10~15 times of the number of covariables.
Comment 6: Problems about result: relationship between clarithromycin resistance and Endoscopic findings in univariate and multivariate analyses.
Response: In Table 5, clarithromycin resistance is associated with Endoscopic findings (P: 0.017, OR: 0.455, 95%CI: 0.235-0.880), which is consistent with the results of multivariate analysis.

Reviewer 2 Report
Manuscript titled ‘‘Identification of antibiotic resistant of Helicobacter pylori and related risk factors in Yangzhou, China: A cross-sectional study’’ aimed to investigate the resistance of H. pylori to five commonly used antibiotics and risk factors in Yangzhou, China. The author investigated an important problem that needed continuous assessment and evaluation. However, there is a lack of consistency and coherence between the title of the study, the aim, the methodology, the results, and hence the discussion section. Moreover, the methodology and the results section were not clear and showed major concerns. The following points are the main Major and Minor issues for the authors to address:
Major issues:
- The title of the study is incomplete and does not reflect the aim of the study or the approach of the study. The meaning of antibiotic resistant in the title (Identification of antibiotic resistant of Helicobacter pylori and related risk factors in Yangzhou, China: A cross-sectional study) is not clear whether it is assessing the antibiotic resistant pattern as mentioned in the aim (line 55) or Identification of antibiotic resistant genes as the study used molecular analysis to identify resistance genes in the isolates (line 98).
- The aim of the study is not related to the approach that has been applied as assessing antibiotic resistant pattern means a description of the antibiotic resistance testing results for an isolate. However, the authors did not use antimicrobial susceptibility testing to determine the phenotypic profile of antibiotic resistant.
- In the methods section, the authors stated that susceptibility to 20 clarithromycin, levofloxacin, metronidazole, amoxicillin, and tetracycline were tested using gene 21 chip technology. However, the purpose of using this approach is not clear and is not related to the title and the aim of the study.
- In the methods section, authors mentioned prior reports of detecting several point mutations, and again the objective of this point in the methodology is not related to the main aim of the study and is not reflected in the results and the discussion sections.
- In the methods section, the gene chip detection of the H. pylori technique does not reveal the number of isolates examined using this test, and Table 2 (line 107) was incorrectly presented, with a lack of description and main sub-headings.
- In the methods section, the authors used multivariable logistic regression to investigate potential factors of H. pylori drug resistance. This study, however, is incorrect since Multivariable logistic regression is utilized when there is a single category outcome and more than one independent variable. The categorical outcome should be obvious and evident. The authors consider the isolates' resistance as an outcome, despite the fact that the resistance analysis was done at the gene level, which does not necessarily reflect the real phenotypic characteristics and is not considered an evident outcome of those isolates in terms of antibiotic susceptibility.
- Regarding the results section, the description of Antibiotic resistance patterns of H. pylori is not correct and does not show the standard interpretation of gene chip technology of resistance genes. The results have been presented in frequencies and percentages which seems to be an interpretation of rather an antibiogram data
- The discussion section should be modified to reflect the study's approach and findings.
Minor issues:
- English should be thoroughly revised.
- H. pylori should be always written in italic
- Tables layout should be adjusted

Author Response
Dear Editors and Reviewers:
Thank you for your letter and for the reviewers’comments concerning our manuscript entitled “Identification of antibiotic resistant of Helicobacter pylori and related risk factors in Yangzhou, China: A cross-sectional study” (ID: jcm-2052575). Those comments are all valuable and very helpful for revising and improving our paper, as well as the important guiding significance to our researches. We have studied comments carefully and have made correction which we hope meet with approval. The main corrections in the paper and the responds to the reviewer’s comments are as flowing:
Responds to the reviewer’s comments:
Reviewer 2:
Comment 1: Problem about the title of the study.
Response: Thank you very much for your careful review. Our title is not precise enough, thank you again for your comment, and change the title to “Antibiotic resistance of Helicobacter pylori and related risk factors in Yangzhou, China: A cross-sectional study”, Please give us your valuable advice.
Comment 2: Problem about the method of gene chip detection.
Response: At present, there are many methods to identify the resistance of Helicobacter pylori, such as drug sensitivity test, RT-PCR, gene chip method, DNA sequencing and so on. Compared with the traditional drug sensitivity test such as T-test, molecular biology technology shows the advantages of more convenient, fast and economical. In addition, some studies have proposed that gene chip method, DNA sequencing and drug sensitivity test have similar results in identifying drug resistance of Helicobacter pylori. Therefore, drug resistance genes were identified in this study to determine the drug resistance of the bacteria. We believe that this simpler and more economical method is more suitable for our study, because it can be carried out in mass screening, so that we can better understand the resistance situation of Helicobacter pylori in the local area, and finally guide clinical treatment.
References:
- Wang YH, Li Z, Wang L, et al. A systematic review and meta-analysis of genotypic methods for detecting antibiotic resistance in Helicobacter pylori. Helicobacter. 2018;23(2): e12467.
- Yin G, Bie S, Gu H, et al. Application of gene chip technology in the diagnostic and drug resistance detection of Helicobacter pylori in children. J Gastroenterol Hepatol. 2020;35(8):1331-1339.
- Hofreuter D, Behrendt J, Franz A, et al. Antimicrobial resistance of Helicobacter pylori in an eastern German region. Helicobacter. 2021;26(1): e12765.
- Zhang J, Zhong J, Ding J, et al. Simultaneous detection of human CYP2C19 polymorphisms and antibiotic resistance of Helicobacter pylori using a personalised diagnosis kit. J Glob Antimicrob Resist. 2018; 13:174-179.
Comment 3: Problem about Table 2 (the original version).
Response: Thank you very much for your careful review. This table was actually formatted during the file conversion process, and we reformatted it as a figure.
Comment 4: Problem about the method: multivariable logistic regression.
Response:
(1) We used multivariable logistic regression to investigate potential factors of H. pylori drug resistance, which is shown in Table 6. Antibiotic resistance (CLR, LVX, MTZ, any antibiotic resistance) was the dependent variable, sex, age and independent variables with a P < 0.2 in univariate analysis were included in the multivariate analysis.
(2) Current studies believe that bacterial drug resistance is caused by gene mutation. The mutation sites detected in this study have been reported by many studies to be closely related to bacterial drug resistance, and have been compared with drug sensitivity tests and gene sequencing. But unfortunately we really haven't been able to find all the sites that cause bacterial resistance.
References:
- Fontana C, Favaro M, Minelli S, et al. New site of modification of 23S rRNA associated with clarithromycin resistance of Helicobacter pylori clinical isolates. Antimicrob Agents Chemother. 2002;46(12):3765-3769.
- Bogaerts P, Berhin C, Nizet H, Glupczynski Y. Prevalence and mechanisms of resistance to fluoroquinolones in Helicobacter pylori strains from patients living in Belgium. Helicobacter. 2006;11(5):441-445.
- Qureshi NN, Gallaher B, Schiller NL. Evolution of amoxicillin resistance of Helicobacter pylori in vitro: characterization of resistance mechanisms. Microb Drug Resist. 2014;20(6):509-516.
Comment 5: Problems about the results of gene chip technology.
Response: We have made correction according to the Reviewer’s comments. We add Table 2 to illustrate the visual results obtained by the gene-chip technology. It is also convenient for people to have a more intuitive understanding to this technology. In the discussion, we introduced what is the gene chip technology and cited literature showing that it also has good sensitivity and specificity compared to the drug sensitivity test.

Round 2
Reviewer 2 Report
Dear Authors
Thank you for responding to some of my comments; however, for future work, keep the reviewer narrative as is so that it is easier to remember and follow your response.
One of my major points has gone unnoticed:
In the methods section, authors mentioned prior reports of detecting several point mutations, and again the objective of this point in the methodology is not related to the main aim of the study and is not reflected in the results and the discussion sections.
Furthermore, the results of resistance gene chip technology do not conform to the standard interpretation of such a technique.
The authors used multivariable logistic regression in the methods section to investigate potential factors of H. pylori drug resistance. This comment was not correctly interpreted and addressed in the results and discussion sections. The significance value and CI 95% of such analysis should be included for each parameter
